# Refolding and characterization of two G protein-coupled receptors purified from *E. coli* inclusion bodies

**Bastian Heim** [1]*, **René Handrick**[1], **Marcus D. Hartmann**[2], **Hans Kiefer**[1]

**1** Institute of Applied Biotechnology, University of Applied Sciences, Biberach, Germany, **2** Max Planck Institute for Developmental Biology, Tübingen, Germany

* kiefer@hochschule-bc.de

## Abstract

Aiming at streamlining GPCR production from *E. coli* inclusion bodies for structural analysis, we present a generic approach to assess and optimize refolding yield through thermostability analysis. Since commonly used hydrophobic dyes cannot be applied as probes for membrane protein unfolding, we adapted a technique based on reacting cysteines exposed upon thermal denaturation with fluorescent 7-Diethylamino-3-(4-maleimidophenyl)-4-methylcoumarin (CPM). Successful expression, purification and refolding is shown for two G protein-coupled receptors (GPCR), the sphingosine-1-phosphate receptor S1P$_1$, and the orphan receptor GPR3. Refolded receptors were subjected to lipidic cubic phase crystallization screening.

## 1. Introduction

G protein-coupled receptors (GPCRs) are the largest protein family in the human proteome with about 865 members sharing a common architecture of seven transmembrane α-helices [1–3]. In response to a variety of extracellular stimuli they transmit signals to downstream effectors inside the cells, primarily through coupling to heterotrimeric guanidine nucleotide binding proteins (G proteins), β-arrestin and kinases [2]. Major biological and pathological processes in neural, cardiovascular, immune and endocrine systems and in cancer are controlled by GPCR signaling. As such, GPCRs are implicated in a multitude of diseases [2, 3] and are currently the most prominent therapeutic target family mediating the action of more than 40% of clinically approved drugs [3, 4]. For structure-guided drug discovery and to gain insight into the mode of action at molecular level, high-resolution structures of GPCRs in complex with various ligands would be desirable [5]. Unfortunately, despite tremendous efforts, progress in structure analysis of membrane proteins, especially in the GPCR field, has proven to be relatively slow for mainly technical reasons. The latter include low level expression, marginal stability of proteins solubilized in detergent micelles and low propensity to form X-ray diffracting crystals. As of 2019, structures of about 53 unique GPCRs, many of which from non-human species, have become available [6, 7]. Significant progress in the field was made by combining various techniques: engineered GPCR fusion proteins and site-

**Data Availability Statement:** All relevant data are within the paper and its Supporting Information files.

**Funding:** This study was supported by the Cooperative Research Training Group

Pharmaceutical Biotechnology stated by the Postgraduate Scholarships Act of the Ministry for Science Research and Arts of the federal state government of Baden-Württemberg, Germany. The funders had no role in study design, data collection and analysis, decision to publish, or preparation of the manuscript.

**Competing interests:** The authors have declared that no competing interests exist.

directed mutations both improved thermal stability as well as crystallizability. Lipidic cubic phase (LCP) crystallization strongly improved the odds of obtaining diffracting crystals compared to micelle-based crystallization [8]. Finally, expression systems were optimized to yield larger quantities of recombinant GPCRs. Still today, GPCR structure determination is far from being a routine procedure. To get there, expression systems requiring minimal human interference and short cultivation times would be of advantage. *E. coli* has proven to be the single most successful recombinant expression system for the determination of protein structures deposited in the Protein Data Bank (PDB) [9]. However, this is not the case for membrane proteins due to low expression levels of native protein and the common formation of inclusion bodies (IB), where protein is deposited as a non-native aggregate inside the cell. To date, only one GPCR structure in the PDB is derived from protein expressed natively in *E. coli* [10], as well as one solid state NMR structure of a GPCR refolded from *E. coli* inclusion bodies [11].

To streamline production of GPCRs, we explored a generic method based on IB production in *E. coli*, purification by immobilized metal affinity chromatography (IMAC), refolding, and polishing by size exclusion chromatography (SEC). GPCRs were stabilized by replacing the third intracellular loop 3 (IL3) by T4 lysozyme [12]. Refolding was initiated by detergent exchange while bound to the Ni-IMAC column. The main focus of the present work was to optimize thermal stability and refolding yield as required for crystallization. Both were assessed by differential scanning fluorimetry (DSF) using the cysteine-reactive dye 7-Diethylamino-3-(4-maleimidophenyl)-4-methylcoumarin (CPM) [13]. Hydrophobic dyes such as SYPRO Orange, commonly used for DSF of soluble proteins, are not compatible with membrane proteins due to their strong fluorescence in the presence of detergents. CPM fluorescence on the other hand is dependent on its covalent reaction with free cysteines that become exposed upon denaturation and can be used both for soluble and membrane proteins. We selected two GPCRs for our studies. The sphingosine-1-phosphate receptor subtype 1 ($S1P_1$) was chosen as a positive control based on its published X-ray structure in complex with the antagonist W146 [14]. Moreover, the orphan receptor GPR3 was selected as an example of a GPCR with unknown structure. This receptor has been shown to be involved in pain sensation, anxiety and addictive behavior by knockdown studies. It has no known natural ligand, but binds to the synthetic inverse agonist (AF64394). Its structure would be a valuable starting point for drug discovery projects [15]. We carried out crystallization trials in the presence and absence of ligands using lipidic cubic phase (LCP) crystallization [14, 16].

## 2. Materials and methods

### 2.1 Construction of GPR3 and $S1P_1$ expressing clones

*E. coli* Tuner™(DE3) pLacI (Merck Millipore, Merck KGaA, Darmstadt, Germany) cells were transformed with the expression plasmid pET23(d)-$S1P_1$ or pET23(d)-GPR3 and plated on LB-agar plates with 0.5% (v/v) glucose and 120 μg/ml ampicillin. One colony of each transformation was used to prepare overnight cultures which were cultivated at 37°C in 200 ml Terrific Broth (TB, Carl Roth GmbH + Co. KG, Karlsruhe, Germany) media with 120 μg/ml ampicillin, 0.5% glucose (v/v) and 150 rpm with 19 mm shaking motion (New Brunswick innova 4200 incubator shaker). The overnight cultures were used to inoculate a fresh culture with 500 ml TB media with the settings mentioned above. After reaching an $OD_{600\ nm}$ of 2.0 the culture was used to prepare glycerol stocks that were stored at– 80°C until further use. To prepare glycerol stocks the bacterial suspension was centrifuged for 10 min with 5500 x g (Sorvall Evolution RC, Thermo Fisher scientific GmbH, Dreieich, Germany). The bacterial cell pellet was resuspended in sterile TB media supplemented with 50% sterile glycerol (Carl Roth GmbH + Co. KG, Karlsruhe, Germany).

## 2.2 Protein expression and preparation of inclusion bodies

The prepared glycerol stocks were used to inoculate 500 ml TB media with 70 µg/ml ampicillin and 0.5% glucose (v/v). These expression cultures were incubated at 37˚C and 150 rpm until they reached an $OD_{600 nm} \geq 2.0$. Protein expression was induced by adding 2 mM of isopropyl-β-D-thiogalactoside (IPTG) and growth continued for 1 h at 25˚C and 150 rpm. The expression cultures were harvested by centrifugation (10 min at 5500 x g and 4˚C), resuspended in 20 ml PBS (137 mM NaCl; 2.7 mM KCl; 10.1 mM $Na_2HPO_4$; 1.8 mM $KH_2PO_4$; pH 7.5) and stored at– 80˚C. Bacterial pellets were treated with three freeze-thaw cycles and incubated afterwards in the presence of 1 mg/ml lysozyme (Carl Roth) for 1 h at 37˚C. After lysozyme treatment, the bacterial suspension was ultrasonicated for 7 min using a tip sonicator (BANDELIN electronic GmbH & Co. KG, Berlin, Germany) while being kept on ice. Temperature was monitored and did not exceed 20˚C. After sonication, 1% (v/v) Triton-X100 and 5 mM EDTA were added and the sample was incubated for 15 min on a rotary shaker (Multi-Rotator PTR-60 Grant-bio, VWR International GmbH, Darmstadt, Germany). IBs were harvested by centrifugation (Heraeus Multifuge 3SR+, Thermo Fisher scientific GmbH, Dreieich, Germany) for 40 min at 14,500 x g. After this centrifugation step, IB pellets were washed twice with 10 ml PBS pH 7.5 containing 0.5% (v/v) Triton-X100 and 5 mM EDTA. Pellets were stored at– 80˚C until further use.

## 2.3 On-column refolding of $S1P_1$ and GPR3

**2.3.1 Ni-IMAC purification for stability optimization and CPM assay development.**
$S1P_1$ and GPR3 were both expressed in *E. coli* Tuner™ (DE3) pLacI as IB. Both receptors contained a C-terminal histidine tag for purification with Ni-IMAC. The IL3 in both receptors was replaced by a bacteriophage T4 lysozyme domain to facilitate crystallization [17]. Refolding of $S1P_1$ and GPR3 was performed on a Ni-IMAC using His-Trap FF 1 ml column (GE Healthcare Europe GmbH, Freiburg, Germany) and an Äkta purifier system (GE Healthcare Europe GmbH, Freiburg, Germany). IB pellets resulting from 250 ml culture volume were solubilized in 10 ml equilibration buffer (PBS supplemented with 2% (w/v) sodium dodecyl sulphate (SDS), by 2 min ultrasonication on ice. After 40 min centrifugation at 14,500 x g the supernatant was loaded onto a Ni-IMAC column equilibrated with 5 column volumes (CV) equilibration buffer. Bound receptor was washed using 50 CV equilibration buffer followed by 50 CV refolding buffer (PBS pH 7.5; 0.2 mg/ml n-dodecyl-β-D-maltopyranoside (DDM; Anatrace); 0.04 mg/ml cholesteryl hemisuccinate (CHS; Anatrace)). Bound receptor was step-eluted and collected using 10 CV elution buffer (PBS pH 7.5; 0.2 mg/ml DDM; 0.04 mg/ml CHS; 200 mM imidazole).

**2.3.2 Detergent screen using thermal stability assay.** 8 µg of IMAC-purified and concentrated $S1P_1$ or GPR3 was diluted with the respective buffer composition and subjected to the CPM assay. The detergents screen was performed with both receptors. Three different detergents, Fos-Choline 12 and 14 (Anatrace) as well as DDM were tested at 2 x and 4 x their critical micellar concentration (CMC) in PBS (pH 7.5). CMCs were obtained from [18]. We further evaluated the effect of CHS addition at a mass ratio of detergent: CHS = 5:1.

**2.3.3 Ni-IMAC purification for crystallization of $S1P_1$ and GPR3 and ligand binding assay.** For crystallization attempts and ligand binding assay the purification and refolding scheme was modified. The main difference to the previous protocol was the addition of phospholipids to the refolding buffer in the form of mixed micelles and thereby providing an environment more similar to a membrane bilayer [14, 19]. Excess phospholipids were later washed out as in protocols used to purify native GPCRs solubilized from membranes [16, 20, 21]. The receptor was solubilized with a modified equilibration buffer (50 mM HEPES pH 7.5; 2% (w/v) SDS; 100 mM NaCl). After the sample was loaded and unbound proteins were washed out with 10 CV of modified equilibration buffer, receptor refolding was performed with three

refolding buffers in three consecutive steps. Refolding was achieved with 10 CV refolding buffer 1 (50 mM HEPES pH 7.5; 0.9 mg/ml DDM; 0.18 mg/ml CHS; 0.02 mg/ml L-α-phosphatidylcholine (Sigma-Aldrich Chemie GmbH, Munich, Germany); 0.02 mg/ml glycerolipids (Sigma-Aldrich Chemie GmbH); 0.01 mg/ml cholesterol (Sigma-Aldrich Chemie GmbH)). This step was followed by 10 CV refolding buffer 2 (50 mM HEPES pH 7.5; 0.45 mg/ml DDM; 0.09 mg/ml CHS; 0.02 mg/ml L-α-phosphatidylcholine; 0.02 mg/ml glycerolipids; 0.01 mg/ml cholesterol) and finally 10 CV refolding buffer 3 (50 mM HEPES pH 7.5; 0.5 mg/ml DDM; 0.1 mg/ml CHS). Bound receptor was step-eluted and collected using 10 CV elution buffer (25 mM HEPES pH 7.5; 0.5 mg/ml DDM; 0.01 mg/ml CHS; 200 mM imidazole).

## 2.4 Dialysis and ultrafiltration

After Ni-IMAC the eluted protein containing fractions were pooled and concentrated using Vivaspin®20 (GE Healthcare Europe GmbH, Freiburg, Germany) with 10 kDa molecular weight cut off (MWCO). Up to 20 ml of protein were concentrated using several rounds of centrifugation for 15 min at 4000 x g, 24°C. About 3 ml of concentrated protein were dialysed for 16–24 h at 4°C against 4 l of Ni-IMAC elution buffer (without imidazole) in a Slide-A-Lyzer G2 cassette (MWCO 10 kDa; Thermo Fisher scientific GmbH, Dreieich, Germany). The total protein concentration was determined using BCA assay or NanoDrop™ 1000 (Thermo Fisher scientific GmbH, Dreieich, Germany) UV/Vis spectrophotometer at 280 nm.

## 2.5 Size Exclusion Chromatography (SEC) for GPCR crystallization and ligand binding attempts

The Ni-IMAC elution fractions with refolded $S1P_1$ or GPR3 were collected and concentrated using ultrafiltration in vivaspin centrifugal concentrators with varying sizes (Thermo Fisher scientific GmbH, Dreieich, Germany). The monomeric form of the concentrated receptor was isolated with SEC by Superdex 200 pg 16/60 or 26/60 column (GE Healthcare Europe GmbH, Freiburg, Germany). Sample volume was up to 1–2% of the column volume. The SEC column was equilibrated with 1 CV SEC running buffer (50 mM Tris pH 7.5; 150 mM NaCl; 0.5 mg/ml DDM; 0.01 mg/ml CHS). Monomeric receptor eluted within 1 CV SEC running buffer. Samples containing the monomeric receptor were concentrated to 4.6 mg/mL for $S1P_1$ respectively 8 mg/mL for GPR3 and used for lipidic cubic phase (LCP) crystallization and for ligand binding using the CPM assay.

## 2.6 Analytical methods

**2.6.1 SDS-PAGE and western blot.** SDS-polyacrylamide gel electrophoresis (SDS-PAGE) was performed using 12.5% acrylamide SDS gels. Equal volumes (20 μl) of the eluted receptor proteins after Ni-IMAC or SEC were loaded onto the gels, separated by electrophoresis and stained with Coomassie Brilliant Blue G-250 (Carl Roth GmbH + Co. KG, Karlsruhe, Germany). SDS-Gels were imaged employing a Fusion FX imager (Vilber Lourmat Deutschland GmbH, Eberhardzell, Deutschland). A second SDS-Gel with identical sample loading was prepared for western blot analysis. Prior to protein transfer, Whatman filter paper and nitrocellulose membrane (both Carl Roth GmbH + Co. KG, Karlsruhe, Germany) were incubated for at least 15 min with western blot transfer buffer (25 mM Tris, 192 mM glycin, 10% (v/v) methanol). The blot was set up and performed in a Trans-Blot Turbo Transfer System (Bio-Rad Laboratories GmbH, Feldkirchen, Germany) for 15 min at 1.3 A. The membrane was washed twice with TBS (20 mM Tris HCl; 150 mM NaCl) and blocked for 1 h with 5% (w/v) skim milk powder (SUCOFIN®) in TBST (TBS supplemented with 0.1% (v/v) Tween 20) at room temperature. The primary anti-penta-His antibody (mouse; Qiagen AG, Hilden, Germany) was

diluted 1:1000 and incubated in TBST with 5% (w/v) bovine serum albumin (BSA; Santa Cruz Biotechnology) over night at 4°C. The membrane was washed three times with TBST and incubated for 1 h in 5% (w/v) skim milk powder in TBST containing a 1:2000 dilution of the secondary anti-mouse HRP conjugate (Cell Signaling Technology Europe, B.V., Cambridge, UK). The membrane was washed twice with TBST and twice with TBS. For detection with Fusion FX the blot was incubated for 1 min with a premixed HRP solution (Immobilon Western HRP substrate, Merck Millipore KGaA, Darmstadt, Germany).

**2.6.2 Thermal stability assay (CPM assay).** The thermal stability assay with the CPM dye (Sigma-Aldrich Chemie GmbH, Munich, Germany) was performed according to [13]. Briefly, the CPM dye was dissolved in DMSO (Carl Roth GmbH & Co. KG, Karlsruhe, Germany) to prepare a stock solution at 4 mg/ml, which was stored at– 80°C until further use. The stock solution was diluted 1:40 with CPM assay dilution buffer (20 mM HEPES pH 7.5; 200 mM NaCl; 0.025% (w/v) DDM). The CPM assay was prepared in a total volume of 130 µl in a separate 96 well plate. For the thermal denaturation assay (CPM assay) 8 µg of the dialyzed and concentrated $S1P_1$ and GPR3 were used. Proteins were dialyzed against an excess of Ni-IMAC elution buffer without imidazole. Both receptors were diluted to 120 µl with the respective refolding buffer. Prior to the measurement 10 µl of the diluted dye solution were added, thoroughly mixed and incubated for at least 5 min at room temperature to allow equilibration of the protein with the buffer components. After incubation 100 µl of each sample was transferred in a FrameStar® 480/96 plate (Roche). The CPM assay was performed on a LightCycler® 480 (Roche Diagnostics GmbH, Mannheim, Germany) with a linear temperature increase of 0.5°C per minute between 20°C and 90°C and recording of fluorescence at 450 nm excitation / 500 nm emission. The apparent $T_m$ was derived from a Boltzmann-Fit (y = $\frac{A_1-A_2}{1+e^{(x-x_0)/dx}}$) using the Origin® 2017 software (OriginLab Corporation, Northampton, USA).

**2.6.3 Mass spectrometry for protein identification.** $S1P_1$ and GPR3 were both cut out from Coomassie stained SDS-gels. Sample preparation, in-gel digest and MS/MS analysis were performed by University of Hohenheim (Modul 1 Mass Spectrometry Unit) on a Q Exactive hybrid quadrupole orbitrap MS (Thermo Fisher). Proteome Discoverer and the tandem MS data analysis software (both Thermo Fisher) were used as search engine for protein identification against the internal sequence database of the University of Hohenheim and the theoretical protein sequence of both receptors.

## 2.7 Crystallization in lipidic cubic mesophase and diffraction measurement

Crystallization screens were performed according to [22]. Concentrated receptors were supplemented with 10 µM W146 respectively 5 µM AF64394 (Takeda Pharmaceutical, Tokio, Japan) of the respective ligand and mixed with molten monoolein (Anatrace Inc., Maumee, USA) at a ratio of 40% protein solution to 60% lipid at 20°C. The cubic mesophase was reached through mixing both solutions in coupled microliter syringes (Hamilton) for at least 100 times using an LCP mixer (SPT Labtech Ltd., Melbourne, UK). The cubic mesophase was transferred into a Hamilton syringe and clamped into a mosquito LCP (SPT Labtech Ltd., Melbourne, UK). With the mosquito LCP drops with 50 nl cubic mesophase were placed on a 96-well glass crystallization plate (Molecular Dimensions, Ltd., Suffolk, UK). These drops were overlaid with 800 nl precipitant solution. All crystallization screens employed in the crystallization attempts are summarized in S1 Table. The crystallization plates were covered with a thin glass plate and incubated at 20°C in a rock imager 182 (FORMULATRIX, Bedford, USA). At regular intervals, pictures of the individual experiments were taken. Identified potential crystals were harvested according to [23] and flash-frozen in liquid nitrogen. Diffraction experiments were performed at 100 K and a wavelength of 1 Å on a PILATUS 6M-F detector at beamline X10SA

(PXII) of the Swiss Light Source (PSI, Villigen, Switzerland), using oscillation angles of several degrees from at least two orientations per crystal.

## 3. Result

### 3.1 Purification and refolding of recombinant expressed GPCR

After solubilization of IBs by sonication in SDS, $S1P_1$ and GPR3 were purified and simultaneously refolded on a Ni-IMAC column as described in the method section. IBs of $S1P_1$ and GPR3 were prepared as described and both receptors were purified by IMAC. Purity of the elution fractions was monitored by SDS-PAGE and verified by western blot (Fig 1).

In the elution fractions of GPR3 and $S1P_1$ a distinct band representing the monomeric form of both receptors can be seen. The apparent molecular weight (MW) appeared to be 5–10 kDa lower than the deduced MW (51.6 kDa for GPR3 and 56.7 kDa for $S1P_1$), which is common for membrane proteins [24]. Protein sequence and integrity of the monomer bands were confirmed by tryptic digest and liquid chromatography MS/MS analysis. For $S1P_1$ the MS/MS analysis resulted in a sequence coverage of 45% for 34 exclusive unique peptides and 48 exclusive unique spectra. The MS/MS data for GPR3 revealed in a sequence coverage of 53% for 71 exclusive unique peptides and 83 exclusive unique spectra. Bands other than the assigned monomer band probably result from proteolytic fragments and aggregates of the GPCRs, as they were all detected by western blot analysis using a primary anti-$(His)_6$ antibody. Monomeric bands were also verified by ESI-MS/MS sequencing.

### 3.2 Establishment of CPM assay

A CPM assay based on covalent reaction of cysteines exposed upon unfolding [13] was established using β-lactoglobulin, as a positive control. Varying amounts of β-lactoglobulin were heated in the presence of the CPM dye under controlled conditions (20°C– 90°C), while

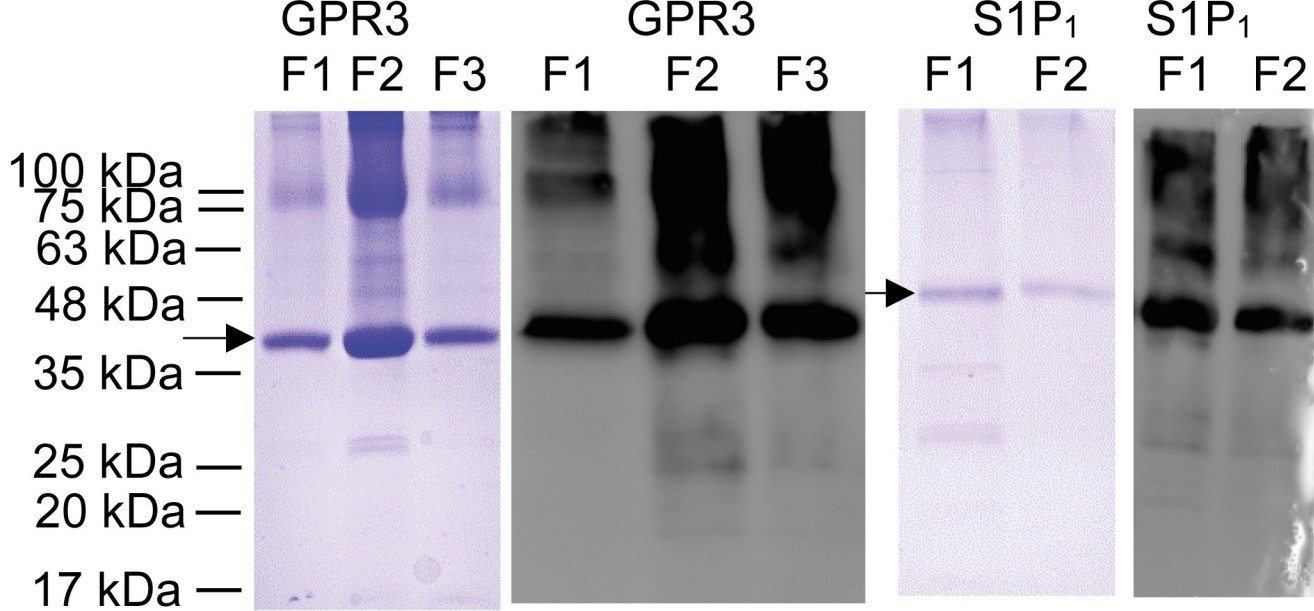

**Fig 1. Ni-IMAC purification of GPR3 and $S1P_1$ analyzed by SDS-PAGE after Coomassie-staining (blue) and verified by western blot (grey/black).** Both proteins were detected with an anti-$(His)_6$ primary antibody (mouse; Qiagen AG, Hilden, Germany) and an anti-mouse HRP labeled secondary antibody. GPR3 F1 –F3: Ni-IMAC elution fraction 1 to 3; $S1P_1$ F1 –F2: Ni-IMAC elution fraction 1 and 2. Bands (monomer) resulting from GPR3 and $S1P_1$ are indicated by arrows.

monitoring fluorescence of the reaction product. An overall sigmoidal fluorescence increase, apparently consisting of at least two transitions, was observed for all samples containing protein, but not for buffer-only and buffer-dye controls. Melting points were determined by fitting curves to the Boltzmann sigmoidal equation (see Methods section). The most consistent results were obtained using 8 μg β-lactoglobulin per experiment. The latter setting was then used to examine assay robustness, reproducibility and to improve the receptor stability. Fig 2 shows as an example the variation obtained in two independent experiments, which were run in triplicate.

The average melting point of these two independent assays was 72.0˚C ± 0.7˚C compared to 75–76˚C as reported by [13]. The unfolding transition displayed a transition in the mentioned two-state model from folded to unfolded protein with small differences in their melting points.

### 3.3 Detergent optimization for GPCR refolding

On column refolding of S1P$_1$ and GPR3 required exchanging the denaturing SDS for a milder detergent or detergent-lipid mixture, providing a more membrane-like environment [14, 25].

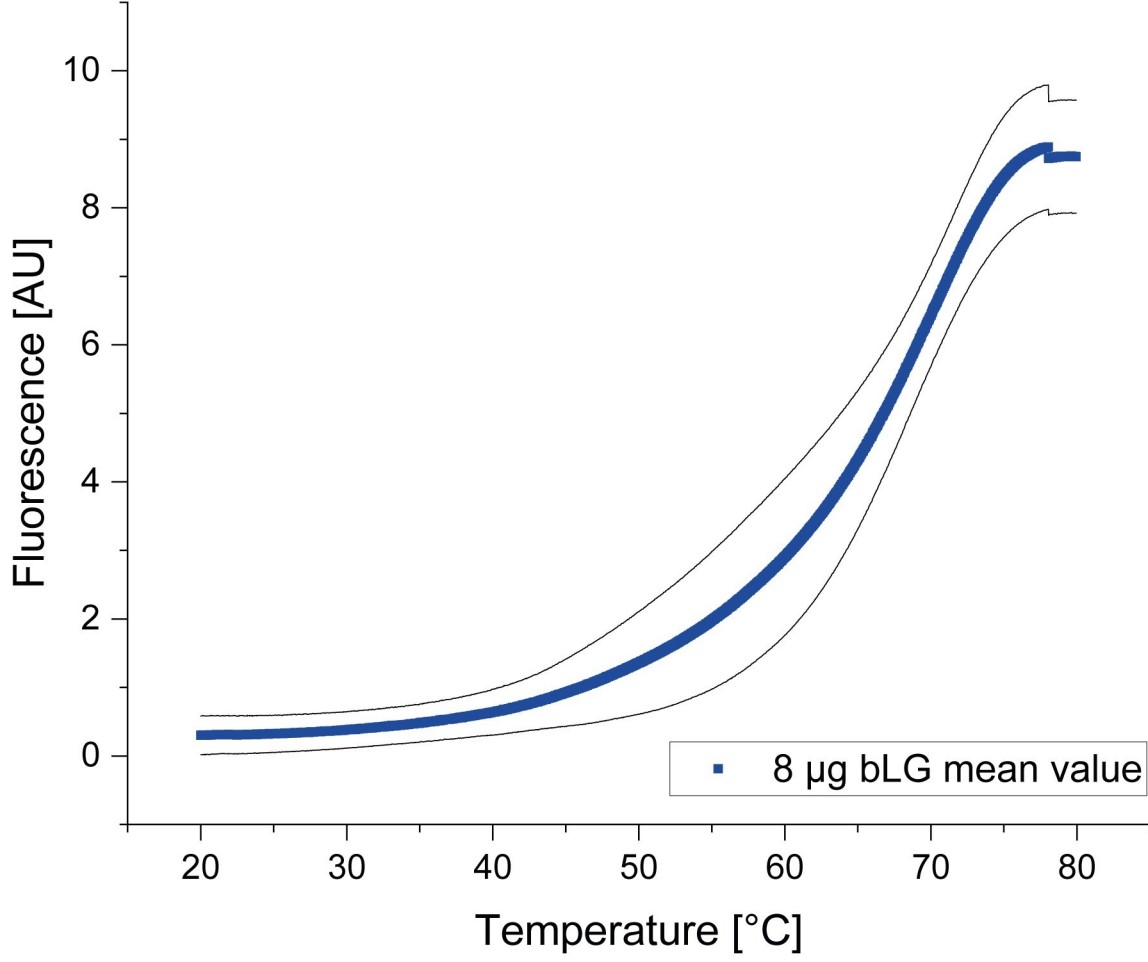

**Fig 2. Unfolding transition of two independent CPM assays performed in triplicates with 8 μg β-lactoglobulin, which was solubilized in CPM assay dilution buffer.** Mean value of unfolding transition of 8 μg β-lactogolbulin (blue) with standard deviation in black. Melting temperatures were determined by fitting each curve to a Boltzmann sigmoidal equation. The average melting point of these two runs was 72.0˚C ± 0.7˚C.

Two zwitterionic detergents (Fos-Choline 12 and 14) as well as the non-ionic detergent DDM and the micelle-forming CHS [26, 27] were included in the optimization. In a first screening, the above mentioned detergents were applied at different concentrations. Receptor stability was measured by CPM assay as described above. As for β-lactoglobulin, the thermal unfolding was measured and a sigmoidal transition was detected. $T_m$ values were determined as above from single Boltzmann fits. Example curves of unfolding transitions for $T_m$ determination are shown for GPR3 in Fig 3.

Apart from Fos-Choline 12 at 2 x CMC, all curves clearly indicate a sigmoidal transition. Addition of CHS shifts the inflection points to higher temperatures when combined with Fos-Choline 12, indicating a stabilizing effect.

Although fluorescence intensity is somewhat variable in replicate experiments, $T_m$ values are sufficiently reproducible. This is shown by low error bars in Fig 4, which summarizes the average $T_m$s of all detergent / GPCR combinations.

GPR3 at 2 x CMC Fos-Choline 14 and CHS did not result in a sigmoidal transition. Hence, no $T_m$ could be determined. Hydrophobicity of GPCRs is variable. 2 x CMC Fos-Choline 14 plus CHS might be a too low concentration to keep GPR3 in solution, resulting in unsuitable refolding conditions. DDM in the presence of CHS resulted in the highest $T_m$ value for both GPCRs, therefore the combination DDM with CHS addition used as a starting condition for further optimization. The DDM concentration with 2 x CMC resulted in the highest $T_m$ values for $S1P_1$ and 4 x CMC for GPR3.

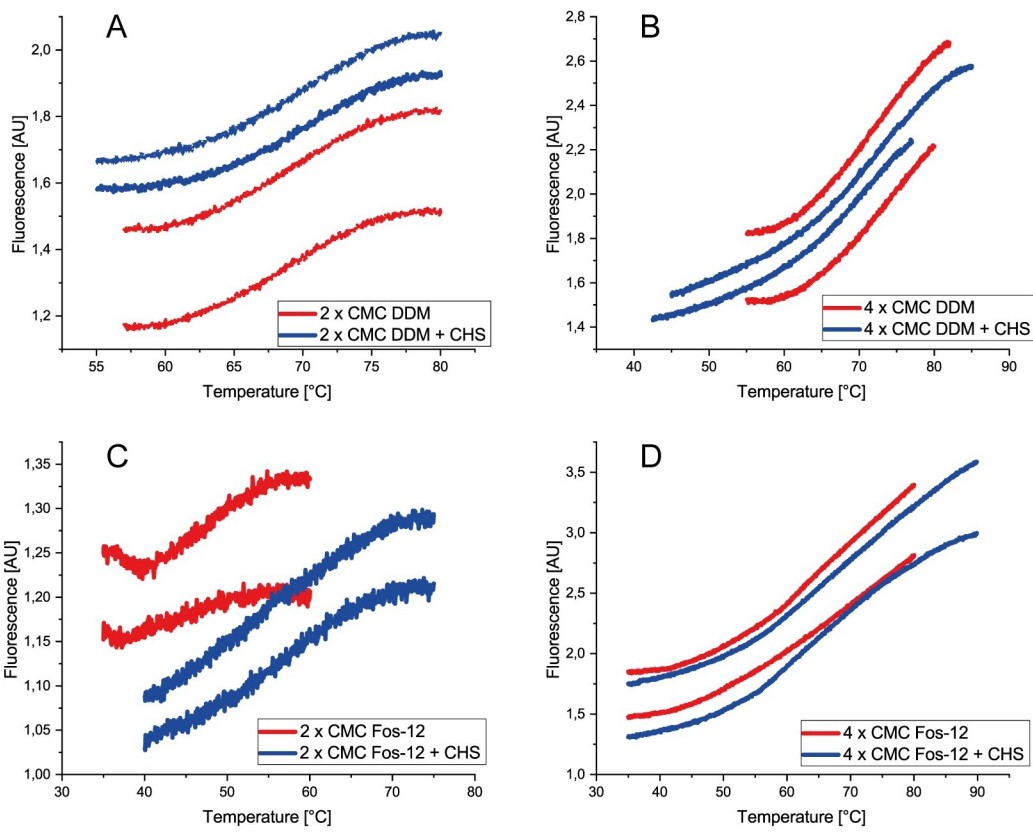

**Fig 3. Unfolding transitions of GPR3 determined by CPM assay, which was performed in duplicates.** A) Unfolding transition of GPR3 in PBS (pH 7.5) with 2 x CMC DDM with and without CHS. B) GPR3 in PBS (pH 7.5) with 4 x CMC DDM with and without CHS. C) GPR3 in PBS (pH 7.5) with 2 x CMC Fos-Choline 12 with and without CHS. D) GPR3 in PBS (pH 7.5) with 4 x CMC Fos-Choline 12 with and without CHS.

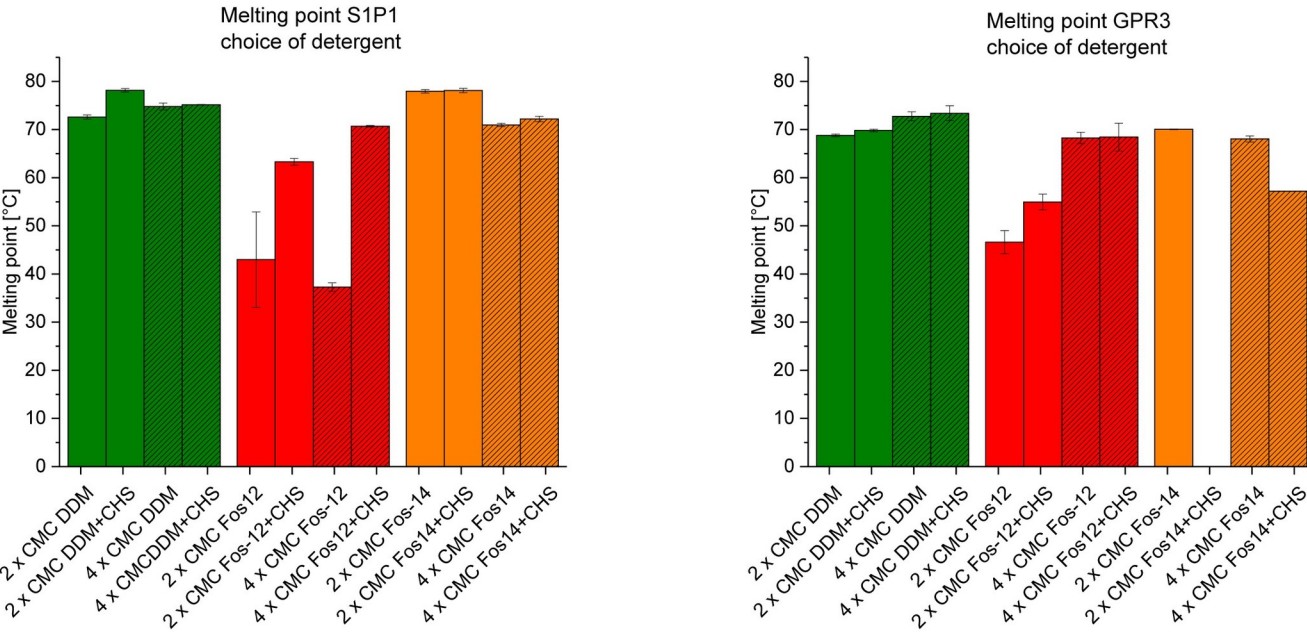

**Fig 4.** Overview of all melting points of S1P$_1$ (A) and GPR3 (B) determined with CPM assay. Green bars display the melting points of DDM with and without CHS, red bars represent the melting points of Fos-Choline 12 with and without CHS and orange bars show the melting points of Fos-Choline 14 with and without CHS. Striped graphs show approaches with four times the CMC of each detergent. Error bars represent standard deviation of duplicates.

**3.3.1 Monomer isolation by SEC and effect of lipid additives on GPCR refolding.** As a second purification step, SEC chromatography on Superdex 200 pg (GE healthcare) was chosen to isolate the full-length GPCR monomers. Subsequent concentration by ultrafiltration resulted in further aggregation. Therefore, we attempted to stabilize GPCRs by adding phospholipids and cholesterol to the detergent micelles to prepare mixed-micelles. This approach was based on the idea that native membrane proteins bind some lipids strongly when solubilized with mild detergents and that these mixed-micelles provide an environment that confers higher stability to the GPCR and prevents precipitation [14, 19, 28]. Those lipids could be important for preserving the native structure. When solubilized from IBs with SDS however, it would be unlikely that those lipids are present in sufficient amounts during refolding. Mixed micelles were produced containing 90% detergent and 10% lipid by mass at various lipid compositions. The isolation of monomers by SEC of S1P$_1$ and GPR3 is shown in Fig 5. The

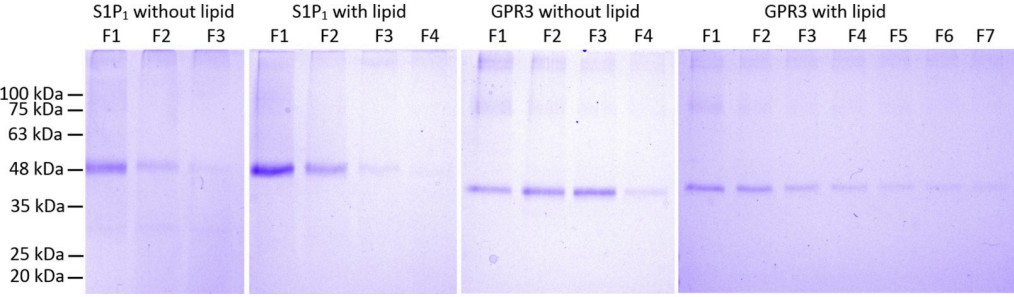

**Fig 5. SEC for the isolation of full-length S1P$_1$ and GPR3 monomers analyzed by SDS-PAGE and Coomassie-staining.** S1P$_1$ and GPR3 monomers after SEC with and without addition of lipids into the SEC running buffer (25 mM HEPES pH 7.5; 500 mM NaCl; 10% (v/v) Glycerol; 0.5 mg/ml DDM; 0.01 mg/ml CHS)) are shown. The samples were loaded after concentration with less than 2% of the total bed volume and run with 1 mL/min. F represents the first SEC fraction with monomer as main compound within the sample.

addition of lipids to the refolding and SEC buffer increased band intensity for $S1P_1$ in the monomer fraction. For both receptors, addition of lipids resulted in an improved separation by SEC and more monomer fractions.

$T_m$s of both receptor derivatives was determined using the CPM assay. The $T_m$ of $S1P_1$ without lipid in the SEC buffer was 51.5˚C ± 0.8˚C while the $T_m$ with lipid in the SEC buffer was 53.2˚C ± 1.8˚C. The $T_m$ of GPR3 without lipid in the SEC buffer was 69.2˚C ± 2.0˚C while the $T_m$ with lipid in the SEC buffer was 71.3˚C ± 0.9˚C. While the $T_m$ of both proteins was not significantly influenced by the addition of lipids, a positive effect of lipid addition on the monomer content and resistance to aggregation upon concentration was evident by SDS-PAGE.

### 3.4 Ligand binding of refolded GPCRs

Although a sigmoidal unfolding transition in CPM assays is a well-accepted indicator of a folded protein, the fluorescence-based CPM assay does not allow calculating the refolding yield. We therefore measured binding of ligand (GPR3 inverse agonist AF64394) to both GPCRs using a shift of $T_m$ as a measure of ligand binding. We were not able to detect a consistent shift in $T_m$ for the ligand AF64394 bound to $S1P_1$. However, for GPR3 the addition of the inverse agonist AF64394 resulted in a significant increase in $T_m$ of about 5.7˚C with a $T_m$ of 72.9˚C ± 0.1˚C with 0 nM AF64394 to 78.6˚C ± 0.3˚C with 10.4 μM AF64394 (Fig 6). Fitting the $T_m$ values of GPR3 with a logistic function resulted in an inflection point of ~ 3 μM AF64394 with a maximum $T_m$ of ~79˚C.

### 3.5 $S1P_1$ and GPR3 crystallization screen using lipidic cubic phase

All crystallization experiments in LCP with $S1P_1$ and GPR3 were performed in presence of the respective ligand. For $S1P_1$ a total of 960 and for GPR3 a total of 1152 crystallization conditions were screened. These screens resulted in the identification of 13 and 6 potential crystals for $S1P_1$ and GPR3, respectively. After elimination of salt crystals identified by X-ray diffraction, two potential non-diffracting $S1P_1$ crystals and two potential GPR3 crystals remained. Fig 7 shows the cubic mesophase in which these four crystals grew. In contrast to the identified salt crystals, these four did not show diffraction patterns indicative for salt crystals.

## 4. Discussion

For crystallization and X-ray structure analysis of GPCRs, sufficient amounts of recombinant protein, conformational homogeneity, high purity and monodispersity are prerequisites. We show here the successful production of two unrelated GPCRs in *E. coli* IBs followed by *in vitro* refolding and chromatographic purification. Compared to eukaryotic expression systems this approach is much faster, less costly and easily yields milligram quantities. However, it also requires optimization of refolding yields and thorough removal of aggregates and degradation products, both of which we addressed. The refolding yield was optimized by introducing a generic thermostability test, while GPCR monomer was purified to near homogeneity using chromatographic techniques. After solubilizing inclusion bodies in SDS, histidine-tagged GPCRs $S1P_1$ and GPR3 were purified in denatured state by Ni-IMAC and refolded by detergent exchange before elution. Analysis of purified proteins by SDS-PAGE (Fig 1) revealed multiple bands, all of which represented GPCR-derived species as verified by Western blot. Beside the respective monomers, SDS-stable dimers and higher aggregates as well as degradation products were observed, similarly to previous findings using the same expressions system [29]. Prior to crystallization, most of the aggregates and degradation products could be removed by preparative SEC, yielding monomeric protein with high purity and monodispersity (Fig 5). To

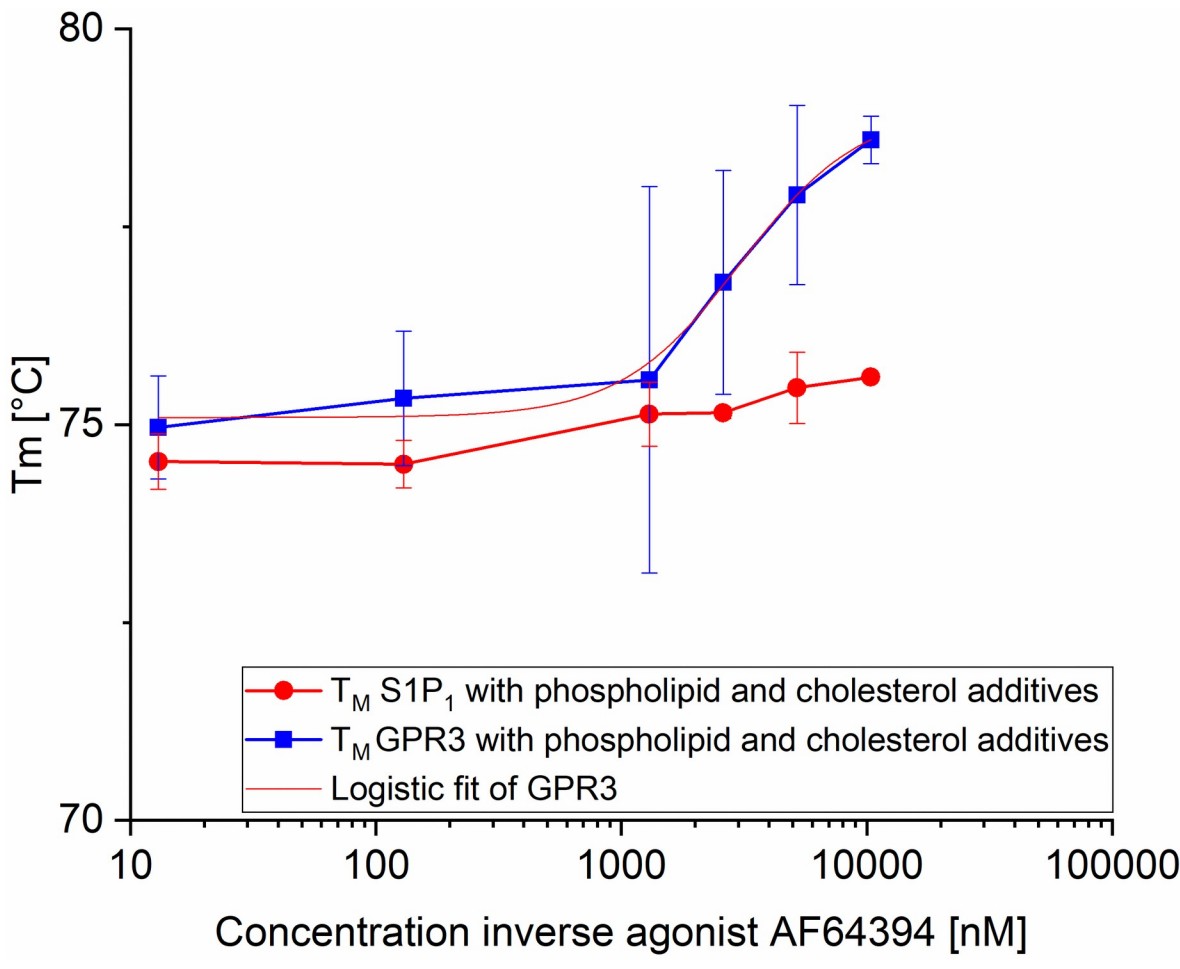

**Fig 6. $T_m$ as a function of ligand concentration.** $T_m$ was measured by CPM assay for SEC purified S1P$_1$ *(control)* and GPR3 in the presence of the of the inverse GPR3 agonist AF64394. Monomers of both receptor derivatives were isolated in buffer composed of 50 mM Tris pH 7.5; 150 mM NaCl; 0.5 mg/ml DDM; 0.01 mg/ml CHS. The values of GPR3 were fitted with a logistic function to determine the inflection point.

optimize refolding yields, we established a thermal shift assay based on a previously described method exploiting the reaction of CPM with cysteines [13]. Initially, this method was validated for the soluble protein β-lactoglobulin as in the original publication. The determined $T_m$ was 72.0˚C ± 0.7˚C, slightly lower than the reported value of 75˚C [30]. More importantly, the low standard deviation of the replicates indicated that the assay was robust and reproducible. Next, two GPCRs were subjected to the CPM assay, namely S1P$_1$ whose structure is known [30], as well as the orphan receptor GPR3 with so far unknown structure [15]. Both receptors were refolded while bound to the Ni-IMAC resin by exchanging SDS for detergent mixtures containing mild detergents [29, 31–33]. The CPM assay yielded sigmoidal curves indicating a cooperative unfolding transition. The assay was then used to analyze thermal stability of refolded GPCRs in different detergent mixtures and to investigate the effect of added membrane lipids. We first compared three detergents at two concentration levels each (Fos-Choline 12; Fos Choline 14; DDM) in the presence and absence respectively of CHS. Both S1P$_1$ and GPR3 showed sigmoidal transitions for nearly all conditions, indicating successful refolding. The only exception where no transition could be observed was for GPR3 in Fos-Choline 14 (2 x CMC) and CHS. Possibly, this detergent concentration was not sufficient to prevent the

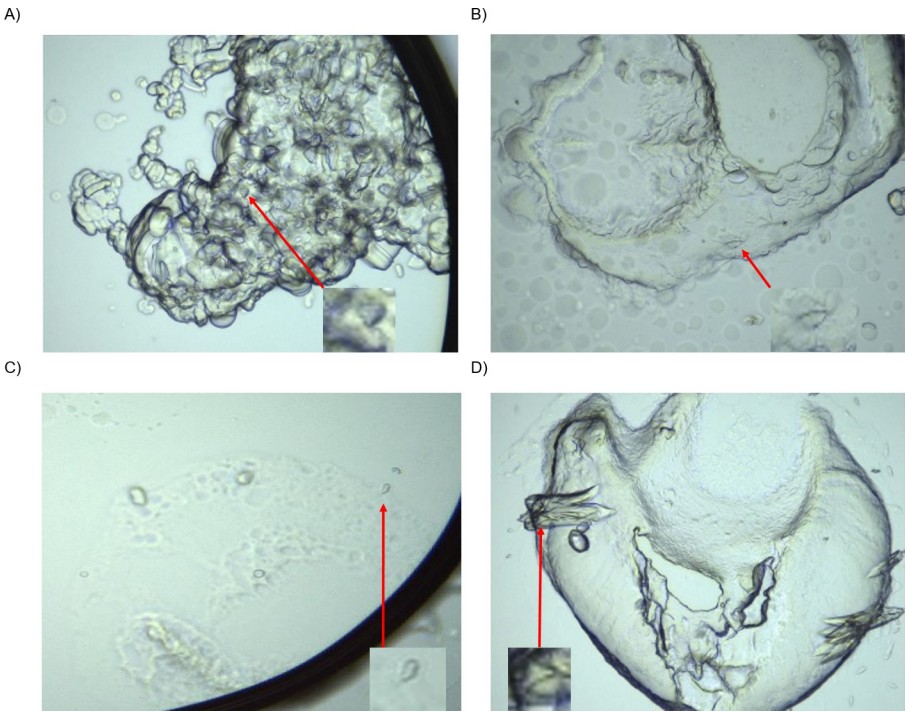

**Fig 7. Microscopic images of the LCP crystallization attempts.** A) S1P$_1$, precipitant: 0.2 M CaCl$_2$, 0.1 M HEPES pH 7.5, 30% (w/v) PEG 4000. B) S1P$_1$, precipitant: 1 M (NH$_4$)$_2$SO$_4$, 0.1 M HEPES pH 7.0 and 0.5% (w/v) PEG 8000. C) GPR3, precipitant: 0.1 M BIS-TRIS pH 5.5 and 2 M (NH$_4$)$_2$SO$_4$ as crystallization condition. D) GPR3, precipitant: 0.1 M 2-(N-morpholino)ethanesulfonic acid (MES) pH 6.5 and 0.6 M Zn(CH$_3$COO)$_2$ as crystallization condition.

receptor from aggregating. Next the apparent T$_m$s were used in a detergent screen to compare different solvent compositions regarding their stabilizing effect and refolding efficiency. DDM and Fos-Choline 14 overall lead to higher T$_m$s and fluorescence intensities than Fos-Choline 12, indicating that longer hydrocarbon chains of those detergents correlate with higher thermal stability. The latter has been observed previously [34, 35]. Due to the zwitterionic headgroup, Fos-Cholines are considered to be stronger detergents than e.g. DDM, which is uncharged [36]. Also, DDM, as a mild detergent, is widely used to solubilize, stabilize and crystallize a variety of membrane proteins, although it might not be ideal to obtain crystals diffracting at high resolution [37]. Due to its broad application and since the T$_m$s of S1P$_1$ and GPR3 only differed marginally when refolded in DDM or Fos-Choline 14, we decided to use DDM for further optimization [36–39]. A slight stabilization resulting from CHS addition was observed both for S1P$_1$ and GPR3. Mixed micelles composed of two or more detergents often form bicelle-like structures more similar to a lipid bilayer than regular micelles and therefore better protect membrane proteins from thermal inactivation [28, 40]. Moreover, the sterol framework of CHS may reduce micelle fluidity similar to cholesterol in lipid membranes and thereby further stabilize GPCRs. Consequently, CHS was added in subsequent experiments. The addition of phospholipids to form mixed detergent/lipid micelles further stabilized both receptors as shown in in Fig 4. In such mixed micelles, lipids may directly interact with the contained membrane protein and stabilize it by providing a more native-like environment [41, 42].

The addition of GPR3 inverse agonist AF64394 to GPR3 showed an increase in T$_m$ by 5.7˚C while a consistent shift of T$_m$ for the ligand AF64394 bound to S1P$_1$ was not observed (Fig 6). The CPM assay does not allow determining the K$_D$ of ligand binding: its low sensitivity

requires a receptor concentration of about 1 μM, preventing the detection of a binding signal at nanomolar ligand concentration where the dissociation constants are expected. Instead, binding will be observed close to the active protein concentration, which is what we find for GPR3 (3 μM). This result therefore not only suggests correct folding through ligand binding activity, but also indicates a high refolding yield, as the concentration of total and active receptor was comparable [43, 44]. At present, we do not know why ligand binding by thermal shift assay was not observed for S1P$_1$.

Results of crystallization screenings are shown in Fig 7. In diffraction experiments, we identified potential protein crystals at four different conditions, two each for S1P$_1$ and for GPR3. Interestingly, for S1P$_1$ those conditions were similar to the ones used in X-ray structure analysis of the same receptor by Hanson and coworkers [15]. Since no diffraction pattern for salt or protein could be observed for these four crystals, but crystals were visually positioned into the beam for diffraction analysis, we assume that these are poorly or non-diffracting protein crystals. Generally, the diffraction from poorly diffracting crystals can be further impaired by the background scatter from adhering mesophase or inhomogeneities within the harvested crystal [45, 46]. The identified crystallization conditions might potentially serve as starting point for further crystallization experiments.

## 5. Conclusion

In this study we were able to overexpress, isolate and refold two human GPCRs (S1P$_1$ and GPR3) expressed in *E. coli* IBs. DDM-CHS mixtures proved to confer the highest thermal stability to the refolded GPCRs compared to other detergent mixtures tested. Stability towards aggregation was further improved by varying solvent composition and addition of phospholipids to mixed-micelles. The same screening and optimization approach may be used for future overexpression and refolding of other GPCRs, allowing to produce scalable amount of native GPCRs using simple and robust techniques. The methods and optimizations established in this study could serve as a generic technique to express, purify and refold GPCRs from *E. coli* IBs. Furthermore, these refolded receptors show promise for crystallization in LCP.

## Supporting information

**S1 Raw images.**
(PDF)

**S1 Table. Summary of all screening solutions used as well as an overview of the varying ingredients used in the crystallization attempts.** Screens marked with a dot were used in crystallization attempts for S1P$_1$ and GPR3. The crystallization screen according to [30] was prepared based on the published crystallization conditions.
(DOCX)

## Acknowledgments

We thank Reinhard Albrecht and the staff of the beamline X10SA at the Swiss Light Source for excellent technical support.

## Author Contributions

**Data curation:** Bastian Heim, Marcus D. Hartmann, Hans Kiefer.

**Funding acquisition:** Hans Kiefer.

**Methodology:** Bastian Heim, Marcus D. Hartmann.

**Project administration:** Bastian Heim, Hans Kiefer.

**Resources:** Bastian Heim, Marcus D. Hartmann, Hans Kiefer.

**Supervision:** Hans Kiefer.

**Writing – original draft:** Bastian Heim, Hans Kiefer.

**Writing – review & editing:** Bastian Heim, René Handrick, Marcus D. Hartmann, Hans Kiefer.

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
