## [Decision Letter · Decision Letter 0]

8 Dec 2020

PONE-D-20-34837

Refolding and characterization of two G protein-coupled receptors purified from E. coli inclusion bodies

PLOS ONE

Dear Dr. Bastian Heim,

Thank you for submitting your manuscript to PLOS ONE. After careful consideration, we feel that it has merit but does not fully meet PLOS ONE’s publication criteria as it currently stands. Therefore, we invite you to submit a revised version of the manuscript that addresses the points raised during the review process.

We look forward to receiving your revised manuscript.

Kind regards,

Sabato D'Auria

Academic Editor

PLOS ONE

Journal Requirements:

"This study was supported by the Cooperative Research

Training Group Pharmaceutical Biotechnology stated by the Postgraduate Scholarships Act of

the Ministry for Science Research and Arts of the federal state government of Baden-

Württemberg, Germany."

"No - The funders had no role in study design, data collection and analysis, decision to publish, or preparation of the manuscript."

Reviewers' comments:

Reviewer's Responses to Questions

**Comments to the Author**

1. Is the manuscript technically sound, and do the data support the conclusions?

Reviewer #1: Yes

Reviewer #2: Yes

2. Has the statistical analysis been performed appropriately and rigorously? 

Reviewer #1: N/A

Reviewer #2: I Don't Know

3. Have the authors made all data underlying the findings in their manuscript fully available?

Reviewer #1: No

Reviewer #2: Yes

4. Is the manuscript presented in an intelligible fashion and written in standard English?

Reviewer #1: Yes

Reviewer #2: Yes

5. Review Comments to the Author

Reviewer #1: The manuscript by Bastian Heim et al. entitled “Refolding and characterization of two G protein-coupled receptors purified from E. coli inclusion bodies” describes mainly the crucial steps needed to obtain quickly and easily (GPCR) transmembrane proteins ready to crystallization experiments with the aim to solve their three-dimensional structure by X-ray.

In this study, two GPCR proteins were chosen S1P1 (as a positive control with a known structure) and GPR3 as example of a GPCR with an unknown structure. Both the receptors were cloned and overexpressed in E.Coli cells. From the collected IB pellets, the two receptors were purified by a Ni-IMAC chromatography strategy. The two receptors were also refolded on the column and the refolding process was optimized following the results obtained by a CPM thermal shift assay. CPM is a dye that reacts with free cysteines becoming exposed upon unfolding, and the reaction enhances fluorescence. Several detergent mixtures and micelle forming agents were explored. DDM-CHS mixtures proved to confer the highest thermal stability to the refolded GPCRs compared to other detergent mixtures tested. Stability towards aggregation was further improved by varying the solvent composition and addition of phospholipids to mixed-micelles. By SEC chromatography monomers of GPCR were isolated. The refolded monomers were tested by the ligand AF64394, to evaluate the stability effect and so the binding capability of the purified receptors. The refolded monomers were implied also in a crystallization screen using lipidic cubic phase, obtain four crystals potentially usable for the X-ray diffraction experiments.

The authors conclude that: “The same screening and optimization approach may be used for future overexpression and refolding of other GPCRs, allowing to produce scalable amount of native GPCRs using simple and robust techniques. The methods and optimizations established in this study could serve as a generic technique to express, purify and refold GPCRs from E. coli IBs. Furthermore, these refolded receptors show promise for crystallization in LCP.”

The aim of this work is interesting and the results convincing. The idea reported is remarkable and the paper is well done.

The manuscript in the present form demands a light revision before it can be published, so this reviewer suggests a “minor revision” of the paper.

Major issue:

1) From line 256 to 259, MS/MS analysis and ESI- MS/MS sequencing results were not reported. Please add more details.

Minor issue:

Just a few suggestions to improve the readability of the text helping the reader to better understand the paper.

1) The abstract is not attractive; it should arouse curiosity in the reader.

2) About the materials and methods section, please reorganize in a more appropriate chronological order this section.

3) In line 233 there is a mistake…in Table 1Error! Reference source not found, please remove it.

4) Results, in the 3.1 subsection please add more details about the IB solubilization.

5) Line 442 misspelling of “purifiy”.

Reviewer #2: The manuscript by Bastian Heim et al. entitled “Refolding and characterization of two G protein-coupled receptors purified from E. coli inclusion bodies” describes the purification, under denaturing conditions, of two G protein-coupled receptors (GPCR), the sphingosine-1-phosphate receptor S1P1 and the orphan receptor GPR3 in E. coli inclusion bodies and the refolding by detergent exchange while bound to the immobilized metal affinity chromatography column.

The authors conclude that the same screening and optimization approach may be used for future overexpression and refolding of other GPCRs, allowing to produce scalable amount of native GPCRs using simple and robust techniques. They conclude, moreover, that the methods and optimizations established in this study could serve as a generic technique to express, purifiy and refold GPCRs from E. coli IBs.

In my opinion the reading of the work is very tiring, the data are not well organized. I admit this manuscript, but with major revision. I believe that these studies need to be described more succinctly. I would suggest to the authors a complete review.

In particular:

- The introduction is very long.

- All graphics have a low resolution.

- Line 91 to 94. Is one colony sufficient to inoculate 200 ml of medium?

- Line 95. With an optical density of 2 is not already beyond the stationary phase?

- Line 102. Why is the concentration no longer 120ug but 70ug?

- Line 105. Is a single hour of post induction at 25 ° C sufficient to produce the protein?

- Line 110. Even if the solution is on ice maybe seven minutes will heat it too much

- Lines 233 to 234. To check

- Lines 256 to 258. The black spots beyond the monomer obtained with the western, why are they all recognized if the antibody is specific against histidine? The bands with molecular weight lower than the monomer could be proteolytic cuts but the aggregates should not exist in denaturing conditions.

- The figures are not in order.

- The figure 3. The colors of the curves in the legends are not seen.

- In figure 3 the curve of 4 x CMC Fos-Choline 14 was not shown.

- Figure 3. The red curves in Graph A do not look similar. I would suggest to describe better.

- Line 301. How do the authors explain the difference?

- Line 302. DDM at 4 x CMC in the presence of CHS resulted in the highest Tm value for GPR3. better for S1P1 DDM at 2 x CMC in the presence of CHS?

- Line 315 to 318. Did the authors quantify the protein? For S1p1 the band seems more evident but for GPR3 no.

- The discussion paragraph is unconvincing, many unresolved points and many assumptions

6. PLOS authors have the option to publish the peer review history of their article (what does this mean?). If published, this will include your full peer review and any attached files.

Reviewer #1: No

Reviewer #2: No

---

## [Author Response · Author response to Decision Letter 0]

9 Feb 2021

Dear Dr. D'Auria,

Thank you for sending the reviewers' comments, which we found very useful to prepare an improved version of our manuscript. Below, we are listing the comments one by one and refer to the changes made in the manuscript where applicable. In addition, the manuscript was updated to the PLOS ONE’s style, the project funding information was removed from the acknowledgements and included in the cover letter. Direct billing is taken over by the Institute of Applied Biotechnology (University of Applied Sciences, Biberach, Germany). The ORCID was created by the corresponding author and is as follows: https://orcid.org/0000-0001-7731-4356. The original uncropped and unadjusted images underlying blot and gel results are attached in the supporting information. We sincerely hope that the revised manuscript is now suitable for publication in PLOS ONE.

With best regards,

Dr. Bastian Heim

Reviewers' comments:

Reviewer #1: 

"(…summary of content)

The aim of this work is interesting and the results convincing. The idea reported is remarkable and the paper is well done.

The manuscript in the present form demands a light revision before it can be published, so this reviewer suggests a “minor revision” of the paper.

Major issue:

1) From line 256 to 259, MS/MS analysis and ESI- MS/MS sequencing results were not reported. Please add more details."

The results for MS/MS analysis have been added (L278 – L281):

"For S1P1 the MS/MS analysis resulted in a sequence coverage of 45 % for 34 exclusive unique peptides and 48 exclusive unique spectra. The MS/MS data for GPR3 revealed in a sequence coverage of 53 % for 71 exclusive unique peptides and 83 exclusive unique spectra."

Minor issue:

Just a few suggestions to improve the readability of the text helping the reader to better understand the paper.

1) The abstract is not attractive; it should arouse curiosity in the reader.

The abstract has been rewritten putting focus on refolding GPCRs for structure analysis.

2) About the materials and methods section, please reorganize in a more appropriate chronological order this section.

The section Material and Methods has been re-organized to bundle expression and purification, refolding, analytical methods and crystallization. Methods for mass spectrometry have been added to the analytics section:

"2.6.3 Mass spectrometry for protein identification

S1P1 and GPR3 were both cut out from Coomassie stained SDS-gels. Sample preparation, in-gel digest and MS/MS analysis were performed by University of Hohenheim (Modul 1 Mass Spectrometry Unit) on a Q Exactive hybrid quadrupole orbitrap MS (Thermo Fisher). Proteome Discoverer and the tandem MS data analysis software (both Thermo Fisher) were used as search engine for protein identification against the internal sequence database of the University of Hohenheim and the theoretical protein sequence of both receptors."

3) In line 233 there is a mistake…in Table 1Error! Reference source not found, please remove it.

Amended, the link within the file to Table 1 has been removed. (L252)

4) Results, in the 3.1 subsection please add more details about the IB solubilization.

"by sonication in SDS" has been added, and there is a reference to the methods section (L263).

5) Line 442 misspelling of “purifiy”. 

Amended. (L491)

Reviewer #2: (…summary)

In my opinion the reading of the work is very tiring, the data are not well organized. I admit this manuscript, but with major revision. I believe that these studies need to be described more succinctly. I would suggest to the authors a complete review.

Abstract, introduction and discussion have been completely revised. In addition, the material and methods section has been reorganized.

In particular:

- The introduction is very long.

This section has been shortened by ~25%. The focus is now entirely on GPCR production, stabilization and crystallization. Functional aspects of the GPCRs have been largely removed.

- All graphics have a low resolution.

Figures have been improved to 300 ppi resolution throughout and were processed by the PACE software. They should now fulfill all requirements detailed in author instructions.

- Line 91 to 94. Is one colony sufficient to inoculate 200 ml of medium?

This procedure was used consistently and never lead to problems regarding glycerol stocks 

- Line 95. With an optical density of 2 is not already beyond the stationary phase?

It was at the end of the logarithmic phase and consistently resulted in viable glycerol stocks. 

- Line 102. Why is the concentration no longer 120ug but 70ug?

The lower concentration was sufficient to keep cultures stable during production. The higher concentration included a safety margin for the glycerol stocks.

- Line 105. Is a single hour of post induction at 25 ° C sufficient to produce the protein?

The expression time had been optimized to yield the highest possible expression level at harvest (not detailed in manuscript).

- Line 110. Even if the solution is on ice maybe seven minutes will heat it too much

Temperature was controlled to not exceed 20 °C. A note has been added in the manuscript (L109). 

"Temperature was monitored and did not exceed 20 °C."

- Lines 233 to 234. To check

Unclear to us what is meant here.

- Lines 256 to 258. The black spots beyond the monomer obtained with the western, why are they all recognized if the antibody is specific against histidine? The bands with molecular weight lower than the monomer could be proteolytic cuts but the aggregates should not exist in denaturing conditions.

Aggregates of multi-spanning transmembrane proteins such as GPCRs are often resistant to SDS. See e.g. Fig. 2 in Biochimica et Biophysica Acta - Biomembranes, 1818(3), 584–591. 

We now addressed this issues in the discussion (L413 – L417)

- The figures are not in order.

Amended, the paragraph below was moved past figure 3.

"Apart from Fos-Choline 12 at 2 x CMC, all curves clearly indicate a sigmoidal transition. Addition of CHS shifts the inflection points to higher temperatures when combined with Fos-Choline 12, indicating a stabilizing effect."

- The figure 3. The colors of the curves in the legends are not seen.

Amended.

- In figure 3 the curve of 4 x CMC Fos-Choline 14 was not shown.

4 x CMC Fos-Choline 14 and the other conditions for S1P1 were not shown in figure 3 because only wanted to show some example curves. The wording was changed as follows (L316 – L317):

"Example curves of unfolding transitions for Tm determination are shown for GPR3 in Fig 3."

- Figure 3. The red curves in Graph A do not look similar. I would suggest to describe better.

We added information in the text and refer to error bars in Fig. 4, showing that Tm is highly reproducible. (L326 – L333)

"Although fluorescence intensity is somewhat variable in replicate experiments, Tm values are sufficiently reproducible. This is shown by low error bars in Fig 4, which summarizes the average Tms of all detergent / GPCR combinations. "

In the caption to Figure 4 the following has been added: "Error bars represent standard deviation of duplicates."

- Line 301. How do the authors explain the difference? 

Hydrophobicity of GPCRs is variable. 2 x CMC Fos-Choline 14 plus CHS might be a too low concentration to keep GPR3 in solution, resulting in aggregation and loss of structure. The following text was added (L335-L336):

"Hydrophobicity of GPCRs is variable. 2 x CMC Fos-Choline 14 plus CHS might be a too low concentration to keep GPR3 in solution, resulting in unsuitable refolding conditions."

- Line 302. DDM at 4 x CMC in the presence of CHS resulted in the highest Tm value for GPR3. better for S1P1 DDM at 2 x CMC in the presence of CHS?

Text has been changed as follows and now matches our findings (L337 – L340):

"DDM in the presence of CHS resulted in the highest Tm value for both GPCRs, therefore the combination DDM with CHS addition used as a starting condition for further optimization. The DDM concentration with 2 x CMC resulted in the highest Tm values for S1P1 and 4 x CMC for GPR3."

- Line 315 to 318. Did the authors quantify the protein? For S1p1 the band seems more evident but for GPR3 no.

A direct comparison of yields with and without lipid addition was not done. The text has been changed to (L355 – L358): 

"The addition of lipids to the refolding and SEC buffer increased band intensity for S1P1 in the monomer fraction. For both receptors, addition of lipids resulted in an improved separation by SEC and more monomer fractions"

- The discussion paragraph is unconvincing, many unresolved points and many assumptions

The discussion has been rewritten to strengthen our conclusions and for more clarity.

---

## [Editor Report · Decision Letter 1]

11 Feb 2021

Refolding and characterization of two G protein-coupled receptors purified from E. coli inclusion bodies

PONE-D-20-34837R1

Dear Dr. Bastian Heim,

We’re pleased to inform you that your manuscript has been judged scientifically suitable for publication and will be formally accepted for publication once it meets all outstanding technical requirements.

Kind regards,

Sabato D'Auria

Academic Editor

PLOS ONE

---

## [Editor Report · Acceptance letter]

15 Feb 2021

PONE-D-20-34837R1 

Refolding and characterization of two G protein-coupled receptors purified from *E. coli* inclusion bodies 

Dear Dr. Heim:

I'm pleased to inform you that your manuscript has been deemed suitable for publication in PLOS ONE. Congratulations! Your manuscript is now with our production department. 

Kind regards, 

on behalf of

Dr. Sabato D'Auria 

Academic Editor

PLOS ONE